# Self-Supervision and Self-Distillation with Multilayer Feature Contrast for Supervision Collapse in Few-Shot Remote Sensing Scene Classification

**Haonan Zhou** [1,*], **Xiaoping Du** [2] **and Sen Li** [2]

1 Graduate School, Space Engineering University, Beijing 101416, China
2 Space Engineering University, Beijing 101416, China; dxp8600@163.com (X.D.); ls_space@163.com (S.L.)
* Correspondence: zhouhaonan011@163.com

**Abstract:** Although the means of catching remote sensing images are becoming more effective and more abundant, the samples that can be collected in some specific environments can be quite scarce. When there are limited labeled samples, the methods for analyzing remote sensing images for scene classification perform drastically worse. Methods that classify few-shot remote sensing image scenes are often based on meta-learning algorithms for the handling of sparse data. However, this research shows they will be affected by supervision collapse where features in remote sensing images that help with out-of-distribution classes are discarded, which is harmful for the generation of unseen classes and new tasks. In this work, we wish to remind readers of the existence of supervision collapse in scene classification of few-shot remote sensing images and propose a method named SSMR based on multi-layer feature contrast to overcome supervision collapse. First of all, the method makes use of the label information contained in a finite number of samples for supervision and guides self-supervised learning to train the embedding network with supervision generated by multilayer feature contrast. This can prevent features from losing intra-class variation. Intra-class variation is always useful in classifying unseen data. What is more, the multi-layer feature contrast is merged with self-distillation, and the modified self-distillation is used to encourage the embedding network to extract sufficiently general features that transfer better to unseen classes and new domains. We demonstrate that most of the existing few-shot scene classification methods suffer from supervision collapse and that SSMR overcomes supervision collapse well in the experiments on the new dataset we specially designed for examining the problem, with a 2.4–17.2% increase compared to the available methods. Furthermore, we performed a series of ablation experiments to demonstrate how effective and necessary each structure of the proposed method is and to show how different choices in training impact final performance.

**Keywords:** remote sensing image scene classification; supervision collapse; multilayer feature contrast; few-shot learning; meta-learning; self-supervised learning; self-distillation

## 1. Introduction

Scene classification for remote sensing images is an important element of the intelligent interpretation of remote sensing images. Its main purpose is to classify according to the semantic information provided by remote sensing images and provide strong support for the application of various remote sensing technologies. Machine learning plays an indispensable role in the classification of remote sensing image scenes and subsequent image interpretation tasks. In [1], the local binary patterns feature is combined to realize the segmentation of ocean and coastal zones, then the support vector machine is used to classify the scenes. Finally, ship detection is realized. Ref. [2] used a deep neural network (without convolution) and an extreme learning machine (ELM) combined with a compressed domain to extract features and classify ships. The position of the ship is determined according to sea–land segmentation. In [3], a support vector machine is used to classify scenes according

to chlorophyll feature information for algal blooms and the coastal monitoring of algal blooms is realized. When classifying scenes, machine learning often requires artificially extracted features in advance. One of the most popular methods is the bag-of-visual-words (BoVW) model [4]. BoVW extracts the frequency of the visual words corresponding to each image block from the visual dictionary for images which serve as visual descriptors. Visual descriptors are used to facilitate scene classification. This hybrid machine learning method combining linguistic and visual features has excellent performance. Therefore, some works have conducted more in-depth research along this line, such as the work on the pyramid of spatial relations [5]. However, these models all rely on the use of prior knowledge to set the representation of features. The performance of the models is limited by the quality of the artificial prior knowledge obtained from the dataset, and this prior knowledge is often difficult to transfer to other datasets. The methods based on machine learning models have some limitations when applied in practical scenarios. Remote sensing image scene classification has developed rapidly in recent years, mainly due to the introduction of deep learning. By employing deep learning, it is possible to process a huge number of remote sensing image data as well as adapt to complex scenes. Most of the deep learning-based remote sensing image scene classification studies have been implemented using Convolutional Neural Networks (CNN) with supervised training on large-scale labeled remote sensing image datasets. The mainstream CNN models are GoogleNet [6], ResNet [7], and DenseNet [8]. The CNN-based methods used in some image classification tasks benefit from parameter adjustment and optimization methods developed over the years [9] and achieve performance that is close to human level [10]. The remote sensing image scene classification methods based on CNN [11–13] learn high-level semantic representation for scene classification through supervised training with a large number of labeled samples and achieve good results.

At present, many well-made remote sensing datasets can be used for the training and testing of deep neural networks, such as the most popular UC merged dataset [4] and WHU-RS19 dataset [14]. However, in actual remote sensing scenes, it is always difficult to obtain training samples. There are few targets in a single remote sensing image, the sample labeling process is cumbersome, and the number of effective data samples that can be obtained is very small. Using a small number of remote sensing image samples to train a deep learning network, the performance of the network will decline significantly. The issue of scene classification from few-shot remote sensing images needs to be solved urgently.

Some methods use deep neural networks to obtain some prior knowledge from Imagenet [10] and transfer it to the new remote sensing image scene classification task outside Imagenet through fine-tuning [4,15], which requires that the target dataset contains a large number of labeled data. When there are few labeled data or the distribution of data has changed greatly, the effect of transfer learning is poor; few-shot learning is intended to make the model learn through a limited number of labeled samples to obtain a reasonable data representation method. This method does not change the representation of common features between different types of samples and can represent the difference between two types of samples represented simultaneously. Meta-learning is a classical few-shot learning method which can avoid overfitting due to limited training data and deepen the learning progress of the network. A model trained by meta-learning can still complete a classification task through a priori knowledge in the face of new types of classification tasks. The few-shot image problem has been approached with meta-learning in many works in remote sensing images during the past few years. Ref. [16] introduced meta-learning to solve land cover classification with limitations on labeled samples. Ref. [17] applied meta-learning to scene classification with limited labeled remote sensing images and achieved good results. Ref. [18] proposed an incremental learning network DLA matchnet to promote the classification of small numbers of remote sensing scene images. Ref. [19] introduced a self-attention module for feature selection and proposed a network

based on multi-scale fusion, which significantly improved the classification accuracy in few-shot scenarios.

Meta-learning is widely used to solve this problem. However, meta-learning has some disadvantages that are difficult to ignore. First, meta-learning demands labeled samples when designing meta-learning tasks. Without a certain number of labeled samples, the meta-learning method will not work. What is more, supervision collapse will occur in the process of meta-learning [20]. In the process of supervised training of a deep neural network, meta-learning will characterize the training samples, enforce the samples with similar categories close to each other in the feature space, and keep away samples of other categories as far as possible. However, the network will be too eager to learn a general representation in the training samples, reduce the loss in the meta-learning as much as possible, only learn the characteristic information conducive to completing the training task, and discard the information irrelevant to the training task. This will affect the extension of the model to new tasks. When the model is faced with unseen data and new tasks, its test performance will be greatly harmed due to supervision collapse. When dealing with the problem of few-shot learning in remote sensing images, these limited labeled samples often appear in the forms of new classes or new tasks. It is unreasonable to retrain meta-learning for new tasks. Therefore, it is very important to solve supervision collapse in few-shot remote sensing image classification.

Supervision collapse shows that a phenomenon similar to overfitting occurs in the process of meta-learning. We believe that the reason for this phenomenon is that meta-learning extracts too little supervision information from the labels of data and that the supervision information lacks universality. So, when the only supervision we can get is label information, how can we train the network to learn a kind of universal supervision information?

To resolve the problems listed above, we propose a method named SSMR based on multi-layer feature contrast. We believe that the target in the remote sensing image is usually composed of smaller parts, and the feature information in the remote sensing image to be classified may be similar to the local features learned in the training process. SSMR emphasizes the comparison of local features generated by multiple convolution layers in the training process, which promotes the network to learn the general feature representation with intra-class compactness and inter-class dispersion. We also propose a self-distillation method using feature contrast to reduce the dependence on labeled samples and boost the performance of the model. In summary, our contributions are as follows:

1.  We use self-supervised learning to reduce supervision collapse in the process of meta-learning. Through self-supervised learning, the embedding network can learn feature information in the data other than the label information while maintaining the invariance of data augmentation (such as cropping and color offset). Our embedding network captures the category information through self-supervised learning and learns an effective feature representation method. We show that self-supervised (contrastive) learning and meta-learning are mutually beneficial.

2.  In the process of self-supervised learning, we use the multilayer features extracted from remote sensing images in different convolution layers of embedding networks for comparative learning and calculate the distance between the corresponding local features or global features for classification. This contrast learning method has little dependence on the label information of data and enhances the generalizability of the model to new unseen data.

3.  We enhance the self-distillation method with multilayer feature contrast. When the previous generation model constrains and transfers knowledge to the next generation model, it makes use of not only the divergence of data label information but also the divergence of feature information extracted from the convolution layers of the previous generation model and the next-generation model. This design enriches the knowledge that can be transferred by self-distillation in few-shot scenarios and

further reduces the dependence of the training process on a small amount of label information, which easily leads to supervision collapse.

4. We construct a challenging dataset to train and test the proposed method and compare it with some representative methods. The experimental results for the dataset show that our method has superior performance and overcomes supervision collapse. We also conducted a series of ablation additional experiments to verify the effect of each module design in our method.

## 2. Related Work

### 2.1. Few-Shot Image Classification

At present, few-shot learning [21–23] is mainly solved by meta-learning. Meta-learning models can be divided into two types. The optimization-based meta-learning model refers to obtaining an initial parameter or model with a good generalization function with a small number of iterative steps. Therefore, meta-learning can train the deep neural network to initialize better on new tasks. MAML [24] learns a set of excellent model initialization parameters (independent of the target task) through SGD [25] so that it can adapt quickly during testing. MAML is unknown to the model, which can be applied to any specific algorithm that uses SGD to update parameters. Instead of calculating the gradient, Reptile [26] uses a soft method to update the slow weight. CAML [27] divides the parameters of the model into context parameters and shared parameters, which can avoid overfitting on a single task and save memory when using a larger network.

Another class of meta-learning [21,28–30] is the metric-based approach. This approach projects the available specimens to the aimed feature space and uses the distance metric function to metricize the task according to the nearest norm (NN). Several methods [31,32] have used various ResNet structured networks as embedding networks in meta-learning and achieved good performance. Rusu et al. created a network named LEO [33] to learn low-dimensional potential embedding networks to reduce computation and complexity. Gregory Koch [34] proposed a Siamese network to train twin networks to extract features from images in the input network in a supervised way and to drive these features to eventually converge on the same class of images. Matching network [28] proposed a model based on an attention mechanism so that the network can learn quickly and be suitable for meta-learning. Prototypical net [21] measured sample distances to various types of sample prototypes in the embedding space, and the simple structure realized strong performance. Tadam [30] used metric scaling and task conditioning to improve the execution of the methods involving meta-learning. Feat [22] proposed an adaptive method constructed on the Transformer [35] module. It improved the feature extraction stage in metric learning and achieved strong performance in few-shot learning.

### 2.2. Self-Supervised Learning

With the help of pretext tasks, self-supervised learning mines the supervision contained in the data itself from unlabeled data and learns valuable representations for downstream tasks through pretext tasks. Pretext tasks have various forms, such as comparing whether different image patches come from the same image [36], predicting the rotation of the input image [37], coloring the image [38], repairing the fragmentary image [39] according to the surrounding conditions, completing the puzzle with picture blocks [40], etc.

Pretext tasks designed based on contrastive learning can be associated with excellent performance. A model performs contrastive learning between positive and negative sample pairs. In [41,42], the positive and negative pairs came from the same image and model, but took different data augmentation; the positive and negative pairs in contractual predictive coding [43] came from the past and the future. In [44], positive and negative pairs came from the same image and data augmentation but were processed using different models (teachers and students). SimCLR [45] used strong data augmentation to generate a great many negative samples for large-batch training and added a projection head after global average pooling to promote self-supervised learning. However, SimCLR relied heavily

on large-batch training and had difficulty in adapting to realistic scenarios without large numbers of data. MoCo [41] regarded contrastive learning as a process of looking up in a dictionary and used a memory bank to maintain the consistent representation of negative samples. MoCo can attain better performance without large-scale experiments. Some recent work has showed that self-supervised learning could contribute to few-shot learning, and the loss of self-supervised learning was introduced in [46,47]. Self-supervised learning was used for instance discrimination to assist training in [48,49].

### 2.3. Knowledge Distillation

Knowledge distillation [50] can transfer the knowledge learned in one network to another. The structures of the two networks can be the same or different. Knowledge distillation usually trains a selected network as a teacher first and then uses the output of the teacher network and labels to train a network as a student, so it is also called "teacher–student learning". Knowledge distillation can reduce the volume of the network and maintain a performance level close to that of the original network. Two networks with different performances can also be combined through knowledge distillation. Han [51] showed that for a given neural network, reducing the network weight by more than 85% through knowledge distillation would not significantly damage the performance of the neural network. According to the object of distillation, knowledge distillation has many forms. KD [50] realized knowledge transfer by minimizing the difference between teachers' and students' classified output labels. Fitness [52] extracted features from the middle layer of the teacher network to guide the student to learn useful knowledge, which can make the student network deeper and narrower than the teacher network. In [53], knowledge was transferred to students in the form of an attention map. IRG [54] constrained the similarity of multiple samples and proposed to constrain the instance relationship graph of students and teachers.

### 3. Stopping Supervision Collapse: SSMR

To overcome the challenge of classifying a few-shot remote sensing image scene, we need to consider how to endow the model with enough prior knowledge and how to enforce the model to learn better in a limited number of remote sensing images. Comparing self-supervised learning and meta-learning, we can see that learning by self-supervision and meta-learning solve the problem of how to train the model to realize classification with few or even no labeled data from two aspects. Meta-learning pays more attention to a small number of training samples. In self-supervised learning, the samples have no labels. By constantly mining the supervision information contained in the data, the feature extraction ability of the network is strengthened, and the network has a good effect on downstream tasks. Meta-learning and learning by self-supervision and meta-learning are complementary in few-shot problems. Therefore, the SSMR method proposed in this paper combines them and uses self-supervised learning based on multi-layer feature contrast to train an embedding network for meta-learning training. Furthermore, it uses the self-distillation based on multi-layer feature contrast to improve the performance of the method in the scene classification mission of few-shot remote sensing images. By comparing the differences between the features extracted from different layers, the feature information contained in a small number of remote sensing images can also be used as the supervision for model training. We can obtain many pre-trained models as the embedding network from different sources. These models are pre-trained in large natural datasets and have strong performance. This requires that the target dataset has sufficient data to support the fine-tuning of the pre-trained models, which is a condition difficult to satisfy in few-shot remote sensing scenarios. At the same time, we want the embedding network to provide an excellent form of remote sensing image feature representation (metric) in meta-learning, while avoiding the occurrence of supervision collapse. In order to equip the embedding network with this capability, a series of designs are used in our method for training the model. These are not experienced by the pre-trained models. Therefore, we design a new



method instead of using existing pre-trained models. The SSMR method flow diagram is shown in Figure 1.

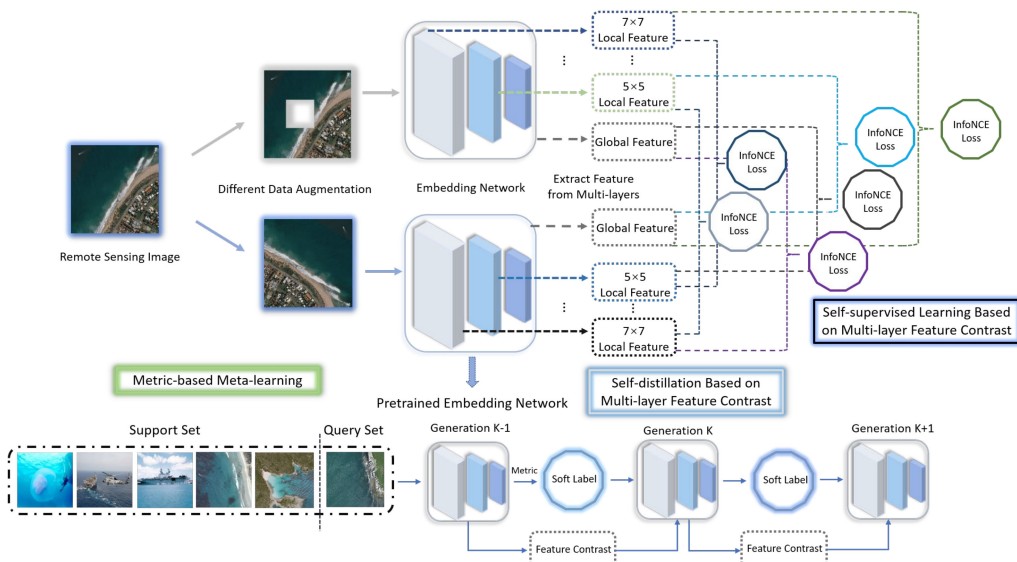

**Figure 1.** The overall architecture of the SSMR method that we propose. The first stage is the self-supervised learning stage, using multi-layer feature contrast. In this stage, a feature extractor is trained as the embedding network of the second stage meta-learning. During the second stage, few-shot learning takes place using meta-learning. In the third stage, the model obtained in the second stage is subject to self-distillation based on multi-layer feature contrast.

### 3.1. Self-Supervised Learning Embedding Network Training Based on Multilayer Feature Contrast

The first step to consider is how to acquire a good neural network embedding network $f_\phi$. If the embedding network $f_\phi$ only obtains the feature information of the seen classes, the subsequent classifier will have difficulty in classifying the unseen classes correctly. The only supervision provided in the few-shot scenarios is a small number of labels. However, unlike the ordinary scene images, remote sensing images have rich texture feature information. Therefore, we choose self-supervised learning that can learn the feature information besides the labels and extract more valuable texture semantic features from the limited labeled input images to better train the embedding network.

Self-supervised learning trains the ability of the embedding network to extract feature information through pretext tasks. We choose the pretext task based on contrastive learning. Different views of a remote sensing image are generated and used as a feed to the embedding network. The embedding network is trained so that the difference between the output of two views from the same source becomes smaller and the difference between the output of views from different sources becomes larger. Finally, the embedding network learns the feature information that can be used for classification. However, compared with high-dimensional data, such as remote sensing images, the effect of using Mean Square Error and Cross-Entropy is not good. If a powerful conditional generation model is used to reconstruct each detail, a lot of computation will be required and the context in remote sensing images will be ignored. Therefore, we choose InfoNCE [43] as the index to measure the mutual information between the comparison objects. InfoNCE compresses high-dimensional data into a more compact hidden space, in which a powerful autoregressive model is used to predict many steps in the future. By calculating the loss function (similar to the method of learning word embedding in the natural language model) through Noise Contrast Estimation, high-level information can be learned from data of multiple modes, such as image, sound, natural language, and reinforcement learning. During the procedure of training, the contrast learning of the embedding network is completed by

guiding the increase of InfoNCE. Training the embedding network in this way helps to suppress overfitting in few-shot tasks.

The second step to consider is how to design a reasonable contrastive pretext task for self-supervised learning. Firstly, if we design the pretext task completely relying on the category label for learning, it can only learn the information related to the category and ignore other more general feature representation information. This defect is particularly obvious in few-shot scenarios. Secondly, when using remote sensing images for comparison, some important targets and scenes in remote sensing images are usually local, and the effect of direct contrastive learning with overall features is not necessarily the best. Especially when the resolution of remote sensing image is high, it is quite tough to obtain good results by directly using global features for contrastive learning; if a remote sensing image is directly represented by a feature vector or local feature map, a lot of information that can be used for classification will be lost, and this loss will be further expanded in few-shot scenarios; it is also difficult to directly use two different remote sensing images for local feature comparison. Even if two remote sensing images belong to the same category, their characteristics in local areas can be extremely different.

Considering the above problems, we adopt self-supervised learning designed as follows: in contrastive learning, the objects used for contrast are taken from multiple views enhanced by different data from the same remote sensing image. Data augmentation is an important method in deep learning, which can significantly enhance the richness of data and the functionality of the deep neural network. According to the characteristics of remote sensing images, we select methods such as random clipping, color jitter, and random gray transformation. Different methods of data augmentation are adopted for the same remote sensing image to expand the differences between different data augmentation results and avoid the collapse of contrastive learning. After obtaining a group of different remote sensing image views with the same source, they are inputted into the embedding network.

To endow the embedding network with the ability to perceive the global features and local features in remote sensing images, we select the feature maps generated by different convolution layers in the embedding network for contrast. The characteristics captured by different layers of the deep neural networks are different. The receptive field of a low-level layer is small and tends to capture local features; high-level layers have larger receptive fields and tend to respond to larger local and even global features. Remote sensing images from the same source will produce a set of similar features, and there will be great differences between the feature sets from different images. At the same time, the features generated by high-level layers come from the features generated by low-level layers. Contrastive learning between them can produce better high-level features. The ones used for contrastive learning may be $5 \times 5$ local features, $7 \times 7$ local features, and feature maps outputted after any convolution layer. Suppose $x_m$ and $x_n$ are the output of the same remote sensing image through data augmentation performed in two different ways. The process of self-supervised learning in this paper maximizes the mutual information between $f_a(x_m), f_5(x_m), f_e(x_m)$ and $f_a(x_n), f_5(x_n), f_e(x_n)$, where $f_a$ represents the global features extracted by the convolution network, $f_5$ represents the $5 \times 5$ local features extracted by the convolution network, and $f_e$ represents the local features of the second layer output of the convolution network. The expression of the InfoNCE loss between $f_a(x_m)$ and $f_5(x_n)$ is:

$$L(f_a(x_m), f_5(x_n)) = \left| log \frac{\exp\{d(f_a(x_m), f_5(x_n))\}}{\sum_{N_x \cup x_n} \exp\{d(f_a(x_m), f_5(x'))\}} \right| \tag{1}$$

where $N_x$ represents the set of all negative samples of remote sensing image $x$ and $d$ represents the square of the Euclidean metric (Euclidean distance). Comparing the output characteristics of different convolution layers will affect the performance of embedding networks. $f_a(x_m), f_5(x_n)$ represents the positive sample pair. $(f_a(x_m), f_5(x'))$ represent the positive sample pair and all negative sample pairs.

The loss of the self-supervised learning stage consists of the InfoNCE loss between local features and local features and the InfoNCE loss between local features and global features. The InfoNCE loss $L(x_m, x_n)$ between $x_m$ and $x_n$ is calculated as:

$$
\begin{aligned}
L(\mathrm{x}_m, \mathrm{x}_n) \quad &= L(f_a(x_m), f_5(x_n)) + L(f_a(x_m), f_e(x_n)) + L(f_5(x_m), f_5(x_n)) \\
&+ L(f_a(x_n), f_5(x_m)) + L(f_a(x_n), f_e(x_m)) + L(f_e(x_m), f_e(x_n))
\end{aligned}
\tag{2}
$$

The first term and the fourth term, the second term and the fifth term are dual relations. Therefore, we set equal coefficients for them. Here, we expect the embedding network to learn a good representation of local features and global features by multilayer feature contrast. The comparisons of global features with local features and of local features with local features at different locations are what we need. Therefore, we will not discuss here which one of them is more important and will assign the same coefficients to them as well. Gather the positive samples continuously from the same remote sensing image, push away total negative samples from other remote sensing images, and integrate the contrast loss of local features and global features into the learning objective. By combining these two losses, we make use of the stability of NCE loss in few-shot scenarios and improve the feature extraction ability of embedding networks when we obtain more transferable embedding networks. In the training process, the probability of the embedding network using the correct category is maximized by continuously reducing $L(x_m, x_n)$ until it is stable at the minimum value. At this time, the self-supervised learning ends, and an embedding network with the ability to extract global and local features of remote sensing images is obtained.

*3.2. Metric-Based Meta-Learning Model Fine-Tuning*

After obtaining an embedding network that has been trained in advance in the self-supervised learning phase, it is embedded in a metric learning framework as an embedding network classifier, which is fine-tuned according to the meta-learning task. Classic meta-learning can be viewed as a multi-task N-way K-shot classification problem [28]. Firstly, N categories (N-way) of probabilistic samples are selected from the overall dataset P, and K + m (m ≥ 1) stochastic remote sensing images are randomly selected from every class. The number of data categories in the overall dataset is greater than N, and the number of instances in each category is greater than K. The actual training and testing sets for meta-learning are composed of individual episodes, and K samples are continuously and randomly selected from the respective sample sets of the N categories as the support set S, which are inputted to the network model for training. The remaining data samples from each class of samples are set together to constitute the query set Q, which is designed to test the capability of the meta-learning model.

In the training stage, all samples in the sample set are inputted to the embedding network to metric, and a metric space is generated. Based on the idea of theoretical mechanics, this paper regards each sample in the metric space as a particle and regards the process of "metric" as the assignment of an impulse to all sample particles by an embedding network. Samples in the category will acquire characteristics after being measured. The centroid of features of this kind of sample in the metric space is obtained as follows:

$$
w_i = \frac{1}{k} \sum_{(x_i, y_i) \in S} f(x_i)
\tag{3}
$$

It will iterate with the updating of samples and calculate the average of each feature. Similar to the concept of the centroid of the particle system, this representation can represent the metric results of the embedding network for this class of samples.

When classifying the test samples, calculate the difference between the samples and each class of sample through the embedding network measurement evaluation results and compare the types with the smallest difference, that is, the class of samples to be classified. Here, the calculation method of the moment of inertia in theoretical mechanics is analogized: each class of a sample set is regarded as a rigid body, and the characteristic

center of each sample is regarded as the momentum at the center of the rigid body. The square ratio of the distance between the measurement result of all samples in each kind and the characteristic center is expressed as the rigid body parallel axis theorem in theoretical mechanics:

$$d(f(q), w_i) = \|wi - f(q)\|^2 \tag{4}$$

Here, we choose Euclidean distance to calculate the distance. Euclidean distance is an effective choice. It is equivalent to a linear model. All the required nonlinearities in our model are learned in the embedding network.

For the type of sample $q$ extracted stochastically from the query set, the softmax prediction is:

$$p(y = i|q) = \frac{\exp(-d(f(q), w_i))}{\sum_{i'} \exp(-d(f(q), w_{i'}))} \tag{5}$$

In the training process of meta-learning classification, softmax loss is the commonly used loss function. However, softmax loss can only continuously increase the distance between categories; it cannot constrain the distance within categories. Therefore, we choose center loss [55], which further minimizes the sum of squares of the distances between the features of the samples in the episode and the feature centers based on softmax loss. The final meta-learning loss function is:

$$L_{meta} = d(f_g(q), wi) + \frac{1}{\lambda} log \sum_k \exp\left(-d(f_g(q), wi)\right) \tag{6}$$

where $\lambda$ is a hyperparameter that is used to adjust the proportion of intra-class compactness and inter-class discrepancy.

### 3.3. Self-Distillation Model Optimization Based on Multi-Layer Feature Contrast

Considering that the features obtained by meta-learning will be over-learned in the training episodes, leading to the occurrence of supervision collapse, we adopt the method of knowledge distillation to simplify the embedding network after self-supervised learning training and meta-learning training, to heighten the generalization ability needed to face new unseen data in the embedding network. Knowledge distillation transfers the knowledge in the teacher model to the student model. Considering that this paper solves the problem in few-shot scenarios, there are not very many labeled data for training the teacher network with good performance. Therefore, we chose self-distillation [56] for model optimization. Since the beginning of distillation training, the structure of the network as a teacher is the same as the structure of the network acting as a student. During training, the output of the teacher network is used as a high-quality soft label to train the student network, which is equivalent to taking the output of the model as the soft training label of the next-generation model. The demand for labeled data is lower than that for the usual knowledge distillation.

Self-distillation iterates the model, and the model of generation K is distilled from the model of generation K-1. The network obtained by self-distillation is expressed by $\varphi'$. In the process of self-distillation, there is a difference loss of prediction label, including the difference between the predictions of the generation K model and ground-truth labels, as well as the difference between the predictions of generation K and generation K-1. These differences are at the label level and are constrained by known data labels and soft labels. Considering the lack of data labels in few-shot scenarios, we introduce feature distillation [52,57]. As shown in Figure 2, we add the difference of feature level to the self-distillation: the difference of the feature extracted from a remote sensing image between the convolution of a layer in the K-generation model and the same layer in the K-1 generation model. The feature distillation can be focal distillation or global distillation, which bridges the gap between the contexts of the adjoining generations.

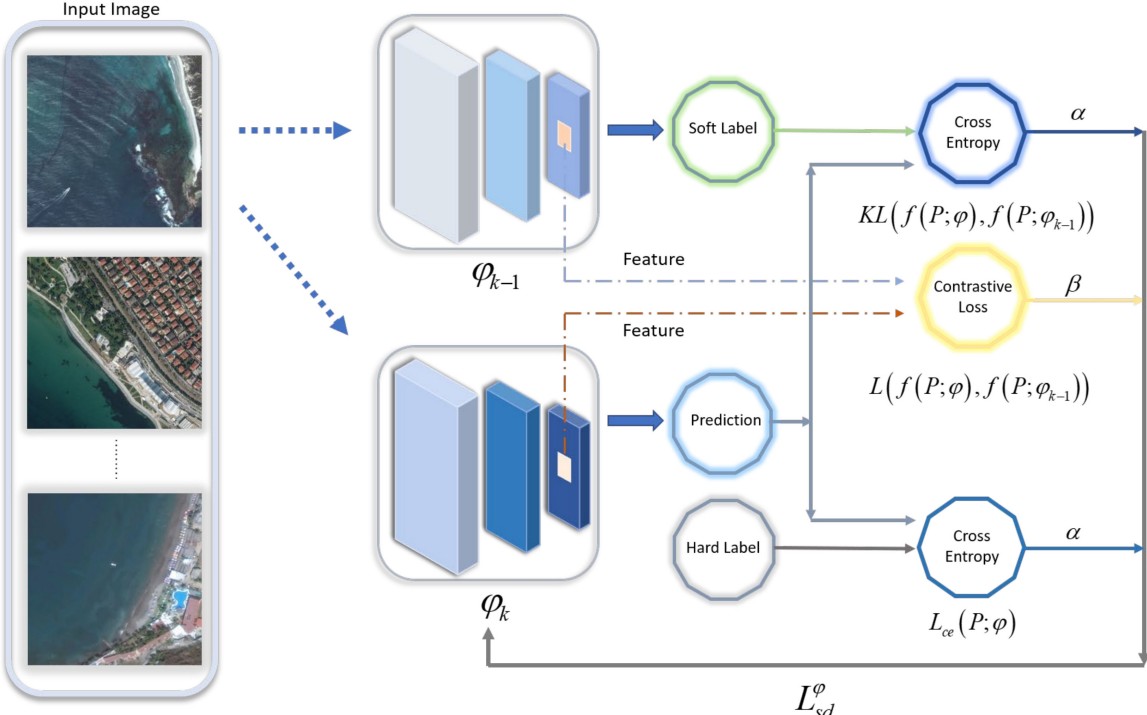

**Figure 2.** The overall architecture of the self-distillation stage. The self-distillation of the model is realized by comprehensively considering the difference between the soft label output by the K-1 model and the prediction of the K-generation model, the difference between the prediction of the K-generation model and the hard label, and the feature difference extracted by the K-1 model and the K-generation model. The selected features can be local or global.

The difference between the predictions of the generation K model and the actual value is measured by cross-entropy loss (contrastive loss), the distinction between the predictions of generation K and generation K-1 is measured by Kullback–Leibler Divergence, and the feature difference extracted from generation K and generation K-1 is measured by InfoNCE. By continuously reducing these differences, the model's classification accuracy is improved. The model that minimizes the sum of these three differences is the K-generation self- distillation model $\varphi_k$. The expression of $\varphi_k$ is:

$$\varphi_k = arg_{\varphi} \min \left( \begin{array}{c} \alpha(L_{ce}(P;\varphi) + KL(f(P;\varphi), f(P;\varphi_{k-1}))) + \\ \beta L(f_{5\times5}(P;\varphi), f_{5\times5}(P;\varphi_{k-1})) \end{array} \right) \tag{7}$$

where $KL$ represents Kullback–Leibler divergence, $L_{ce}$ is on behalf of the cross-entropy loss, $L$ is the InfoNCE loss, $P$ is the overall dataset, and $\alpha$ and $\beta$ are the hyperparameters. The loss function $L_{sd}^{\varphi}$ in the self-distillation process is expressed as:

$$L_{sd}^{\varphi} = \alpha(L_{ce}(P;\varphi) + KL(f(P;\varphi), f(P;\varphi_{k-1}))) + \beta L(f_{5\times5}(P;\varphi), f_{5\times5}(P;\varphi_{k-1})) \tag{8}$$

## 4. Experiment

To start with, we analyze the datasets and model training settings used in the experiment, then make a quantitative comparison with other methods, and finally conduct a series of ablation experiments to verify the effectiveness of various designs.

### 4.1. Dataset

To show the change and diversity of remote sensing image scenes, researchers have constructed more and more large-scale remote sensing datasets that can be used for scene classification, such as the PatternNet dataset [58], the NWPU-RESISC45 dataset [59], and

the RSD46-WHU dataset [60], etc. The NWPU-RESISC45 dataset contains 45 kinds of scenes with 700 remote sensing images per scene, totaling 31,500 remote sensing images. The RSD46-WHU dataset contains 46 kinds. Each kind contains 500–3000 remote sensing images, totaling 117,000 images.

However, directly using the above datasets for meta-learning training cannot approximate actual few-shot remote sensing image application scenes. These datasets contain tens of thousands of remote sensing images, which quantities cannot be obtained in actual few-shot scenarios. At the same time, some remote sensing scene images of different categories are not very different. The training of meta-learning is carried out in the selected episodes. If the network only faces the data in the same or similar domains, this means that there are no data to encourage the network to learn the representation of data with large category differences, which easily leads to supervision collapse.

To avoid that problem and verify whether the SSMR method we created can overcome the supervision collapse, we set up a new dataset. First, the remote sensing images we selected are from the maritime satellite image dataset (MASATI) [61]. The MASATI dataset is compiled from optical remote sensing image scenes in different regions of Europe, Africa, Asia, the Mediterranean, the Atlantic, and the Pacific from March 2016 to June 2019. The dataset contains seven types of samples, including land, coast, ocean, ship, multiple ships, and shore ships. The resolution of each sample is 512 × 512. The MASATI dataset contains a total of 7389 images. Part of the MASATI dataset is shown in Figure 3. There are three remote sensing image scenes, including ships, multiple ships, and shore ships, with similar categories of targets in the dataset. Therefore, the method is required to have the ability to capture local features and global features, and the ability to achieve few-shot classification is also required to be higher.

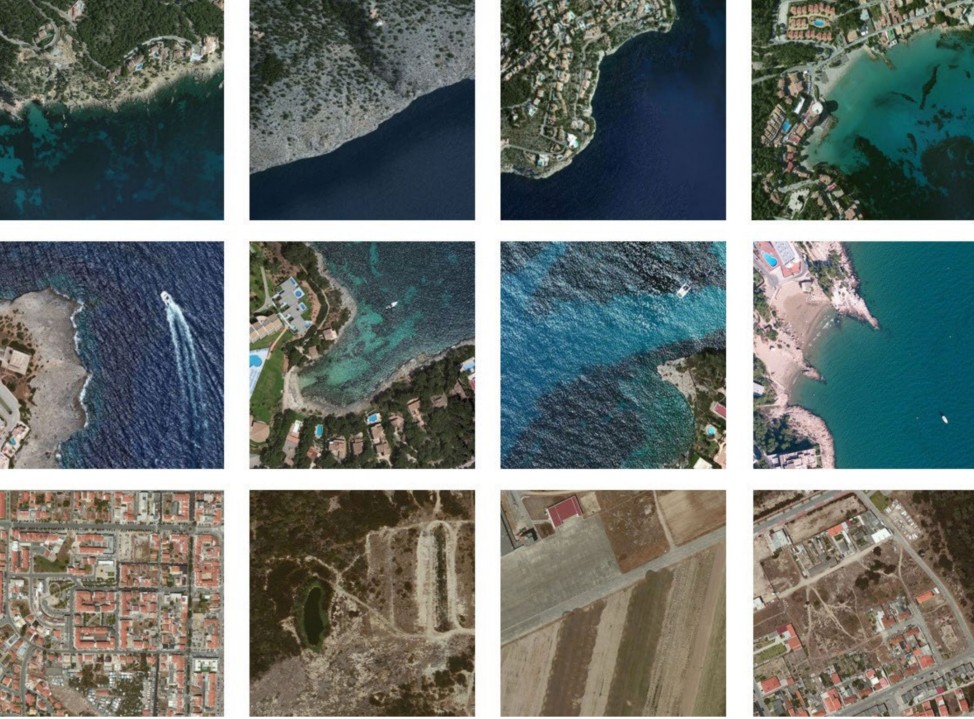

**Figure 3.** The samples in the MASATI remote sensing image dataset have high requirements for the performance of the classification model. Most remote sensing images of marine scenes contain a large number of wave ripples as interference. Especially for several types of data related to ships, the ship itself will be mixed with white ripples, which will increase the difficulty of classification and require a high local feature extraction ability of the model. The network also needs to be able to distinguish single ships, multiple ships, and land ships, which further requires the network to have the ability to capture global features.

We also use data from Miniimagenet [28], a common and famous few-shot dataset. The Miniimagenet dataset is a dataset commonly used to assess the ability of a method designed for few-shot learning. It is a sub-dataset collected from the Imagenet [10] dataset for few-shot research, including 100 conventional image categories, such as birds, dogs, fish, warships, trash cans, with 600 images in each category. As shown in Figure 4, these images are extremely different from remote sensing images in type and structure, making it difficult to directly apply them to the training of a scene classification model for remote sensing images.

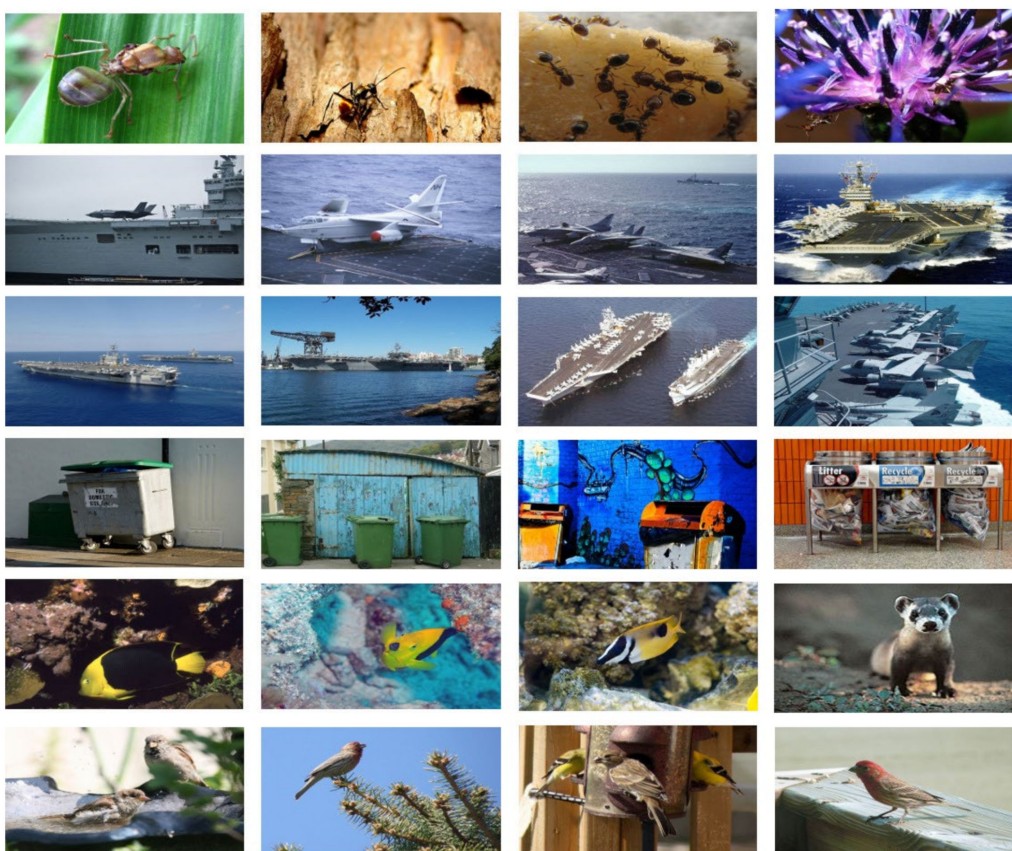

**Figure 4.** Part of the Miniimagenet dataset. The types of data, image structures, and shooting angles are greatly different from remote sensing images.

We mix the MASATI remote sensing image dataset and some data from the Miniimagenet dataset according to a certain proportion, in which the proportion of the MASATI is less than that of the Miniimagenet. We show the composition of the mixed dataset in Table 1. In the episodes selected by meta-learning, there can be large differences between samples. As shown in Figure 5, if the model cannot adapt to the changes between categories, supervision collapse will occur and limit the performance. Such dataset construction can allow training and testing for the cross-domain ability of the network and the ability to deal with unseen data. It can also prevent overfitting by increasing the numbers and types of samples.

**Table 1.** Composition of the mixed dataset we designed, including the categories and number of samples from different datasets. The dataset contains a total of 98 categories, a total of 61,989 samples, of which remote sensing image samples account for 11.9%.

| Samples | Categories | Number |
|---|---|---|
| From MASATI | 7 (land, coast, ocean, ship, multiple ships, etc.) | 7389 (11.9%) |
| From Miniimagenet | 91 (dog, lion, fish, warship, trash can, etc.) | 54,600 (88.1%) |
| Total | 98 | 61,989 |

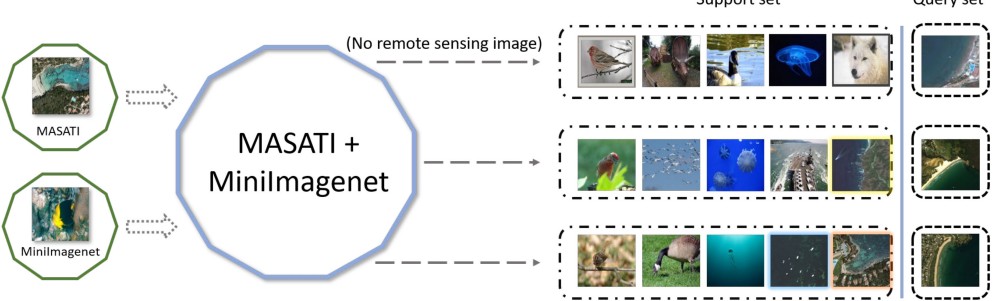

**Figure 5.** We use remote sensing image data in the MASATI and data in the Miniimagenet datasets to create a new dataset for training. The training set contains a small number of remote sensing image data. During meta-learning training, there may be only one or two of the five images sampled in the training set or even no remote sensing images. When testing the performance of the model, the unseen remote sensing images are used to evaluate the cross-data domain ability of the model and whether the methods overcome the supervision collapse.

### 4.2. Model Architecture

Our self-supervised learning network is constructed and adapted based on Augmented Multiscale Deep InfoMax [62]. The network is based on a series of convolutional blocks and residual blocks. Each residual block from ResBlock3 to ResBlock7 contains 16 layers of convolution, where $1 \times 1$ convolutions are used to control the increase of the receptive field and reduce model complexity. The first layer of each block is the average pool with padding of 0, so as not to destroy the distribution station of the features. The kernel, output channels of the kernel, stride, padding, and layer numbers of each layer are shown in Table 2. When we use this network for scene classification, we will only use a FC layer and the following softmax layer as the classifier. However, in this paper we use an embedding network to achieve the representation and metric of remote sensing image features, and it is the distance metric in the meta-learning module that achieves the classification function while avoiding supervision collapse. The embedding network needs to output the feature representation it extracts from the given remote sensing images rather than the scene classification results. Therefore, the FC layer is not used in the embedding network we design. The FC layer is removed from all the embedding networks used in our experiments. At the same time, the Batch Norm [63] causes the data in the support set to interact with information, which eventually results in the images used for training in each episode containing not only the information for the image itself but also the information for other images, further deepening the limitation of information acquisition by the network. Therefore, we use Layer Norm [64] to replace the Batch Norm to avoid the supervision collapse caused by data interacting with information within the same batch.

**Table 2.** Self-supervised learning network structure.

| Layers | Input Size | Output Size | ConvBlocks (Kernel, Output Channels, Stride, Padding, Numbers) |
|---|---|---|---|
| Conv1 | $128 \times 128$ | $62 \times 62$ | $[5 \times 5, 192, 2, 2]$<br>$[3 \times 3, 192, 1, 0]$ |
| ResBlock2 | $62 \times 62$ | $30 \times 30$ | $[4 \times 4, 384, 2, 0]$<br>$[1 \times 1, 384, 1, 0] \times 15$ |
| ResBlock3 | $30 \times 30$ | $14 \times 14$ | $[4 \times 4, 768, 2, 0]$<br>$[1 \times 1, 768, 1, 0] \times 15$ |
| ResBlock4 | $14 \times 14$ | $7 \times 7$ | $[2 \times 2, 1536, 2, 0]$<br>$[1 \times 1, 1536, 1, 0] \times 15$ |
| ResBlock5 | $7 \times 7$ | $5 \times 5$ | $[3 \times 3, 1536, 1, 0]$<br>$[1 \times 1, 1536, 1, 0] \times 15$ |
| ResBlock6 | $5 \times 5$ | $3 \times 3$ | $[3 \times 3, 1536, 1, 0]$<br>$[1 \times 1, 1536, 1, 0] \times 15$ |
| Conv7 | $3 \times 3$ | $1 \times 1$ | $[3 \times 3, 1536, 1, 0]$<br>$[1 \times 1, 1536, 1, 0]$ |

*4.3. Implementation Details*

4.3.1. Data Augmentation

Some recent work [65,66] shows that powerful data augmentation means are really effective in improving performance in self-supervised learning. There are many methods of data augmentation, and different methods have different effects on remote sensing images. We use scale clipping, converting the image into a gray image, adjusting the brightness, contrast, saturation, and color equality, etc. We compare the effects of different methods on the performance of the model through experiments.

4.3.2. Optimizer Selection and Hyperparameter Setting

When training our proposed model, the optimizer has two options. One is the SGD optimizer with a momentum of 0.9, a learning rate of 0.0001, and a weight decay of 0.0005. The other is the Adam [67] optimizer. We choose a learning rate of 0.0002. A total of 200 epochs are trained. We compare the effects of the two optimizers on the function of the model through ablation experiments. When training the model for self-distillation, we use the same optimizer selection strategy and hyperparameter settings and set both $\alpha$ and $\beta$ to 0.5 for a total of 10 epochs.

*4.4. Quantitative Comparison*

In the dataset we set up, the ability of our method is assessed through two commonly used few-shot learning tasks: one-shot and five-shot five-way tasks. We extract samples according to a pre-designed proportion to form episodes as training inputs in the task training stage. Take the one-shot five-way task as an example: an episode includes samples of five classes, each of which contains one support sample for training and 15 query samples for testing. The five-shot five-way task runs similarly. In the testing stage, we evaluate our method in the randomly selected episodes and calculate the average accuracy of multiple tasks with a confidence interval of 95%. At the same time, the performances of some classical methods [21,24,29,68] and recently proposed methods [22,69] are also tested (see Table 3 for more information). Our method achieves an accuracy of $66.52 \pm 0.20\%$. When the number of images received by the model increases, the performance of our SSMR method is significantly improved, and the accuracy in the five-shot five-way task is $83.26 \pm 0.49\%$. This is in line with the nature of self-supervised learning: the more data the model receives, the better it can extract features and the better the effect of self-supervised learning can be. In subsequent experiments, we further increased the number of images received by the

model to observe the change in model performance. ProtoNet uses the pre-trained ResNet as the embedding network and realizes classification through metrics. It is one of the most important methods used in metric learning. RENET [69] is the most advanced method. In the one-shot five-way task, our method achieves 2.4% and 17.2% improvements over RENET and ProtoNet, respectively. Similar accuracy improvements are observed in the five-shot five-way task.

**Table 3.** One-shot five-way and five-shot five-way task accuracy (%) comparison, with 95% confidence.

| Method | Embedding Net | One-Shot Five-Way Accuracy | Five-Shot Five-Way Accuracy |
|---|---|---|---|
| MAML [24] | ConvNet | $53.47 \pm 0.82$ | $66.46 \pm 0.12$ |
| RelationNet [29] | ConvNet | $44.21 \pm 0.14$ | $49.22 + 0.26$ |
| ProtoNet [21] | ResNet18 | $49.27 \pm 0.66$ | $65.16 \pm 0.11$ |
| FEAT [22] | ResNet18 | $47.45 \pm 0.13$ | $57.38 \pm 0.22$ |
| SemiProtoFEAT [22] | ResNet18 | $48.26 \pm 0.22$ | $59.52 + 0.24$ |
| Gcn [68] | Gcn | $46.91 \pm 0.20$ | $59.96 \pm 0.21$ |
| RENET [69] | ResNet18 | $64.11 \pm 0.46$ | $82.32 \pm 0.32$ |
| SSMR (ours) | Self-supervised network | $66.52 \pm 0.20$ | $83.26 \pm 0.49$ |

At the same time, we also note that the performance of FEAT with a standard Transformer structure is not excellent. As a current research hotspot, Transformer has demonstrated strong performance in a variety of tasks. The Transformer mainly consists of mechanisms that emphasize self-attention. Transformer is used in FEAT to implement transformations between different sets. It uses the same embedded network structure as the other methods: ResNet18. FEAT uses the self-attention mechanism for key features extraction and feature transfer between two datasets. On the basis of a comparison with other methods, we argue that: in few-shot problems, the self-attention mechanism motivates the model to pay special attention to the features contained in the training samples and to pass these key features to the new dataset. However, in this process, the self-attention mechanism may abandon the coding of features that are not very useful for distinguishing training samples. The model can only learn the features of existing data, rather than the features that can be generalized in new types of data. Finally, supervision collapse occurs. The method we propose uses a series of designs, such as self-supervised learning, to overcome supervision collapse and achieve excellent performance in tasks.

The small number of samples is the central factor that limits the performance of most methods in few-shot scenarios. To observe the impact of the number of samples on the performance of the model, we provide samples of 10, 15, and 20 shots for several models. The results are shown in Table 4. As expected (Table 2), when the number of samples increases from one to five, the classification accuracy of several models is greatly improved. This shows that with the increase in the number of samples, the feature information extracted by the network becomes rich. However, when we continue to improve shot number, the performance levels of these models are greatly different, as shown in Figure 6. When shot number is raised to 10 and 15, the classification accuracies of most models are not significantly improved. When the shot number is raised to 20, the performance of most models in the test set is still not significantly improved or even decreased. This shows that these methods actually have supervision collapse in the support set (training set), and the learned features are difficult to generalize in the test set. It is not only the small number of samples that restricts the performance of these models, but also the supervision collapse of these models in few-shot learning. The method we have proposed here achieved significant performance improvements when shot numbers were increased to 10, 15, and 20. Our SSMR overcomes supervision collapse; the features extracted from the support set can be generalized in the test set and the classification task can be completed well even in the face of new unseen data.

**Table 4.** Few-shot classification accuracy (%) results for 10 shots on a five-way task, 15 shots on a five-way task, and 20 shots on a five-way task. Ninety-five percent confidence intervals are provided for all results.

| Method | 10-Shot Accuracy | 15-Shot Accuracy | 20-Shot Accuracy |
|---|---|---|---|
| FEAT [22] | $57.95 \pm 0.20$ | $59.06 \pm 0.76$ | $60.71 \pm 0.73$ |
| MAML [24] | $71.04 \pm 0.15$ | $73.93 \pm 0.25$ | $73.44 \pm 0.37$ |
| ProtoNet [21] | $73.18 \pm 0.20$ | $75.57 \pm 0.31$ | $75.46 \pm 0.14$ |
| SSMR (ours) | $85.96 \pm 0.12$ | $86.48 \pm 0.11$ | $88.21 \pm 0.10$ |

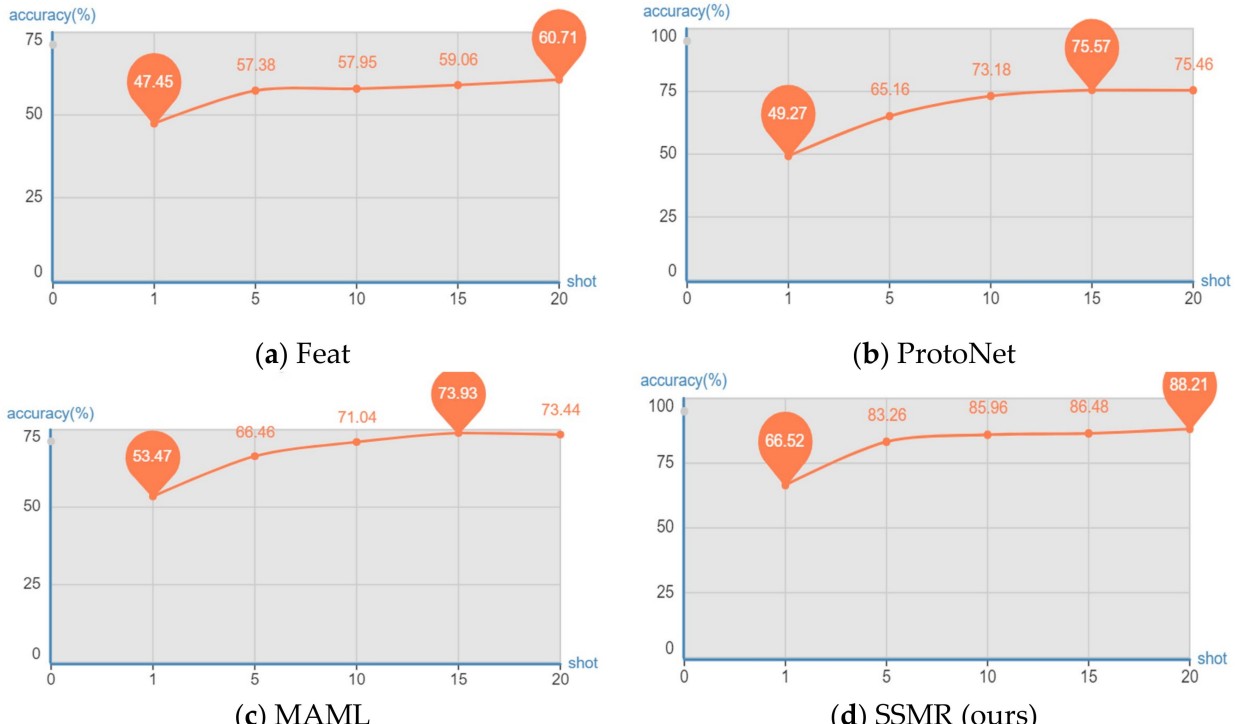

(**a**) Feat

(**b**) ProtoNet

(**c**) MAML

(**d**) SSMR (ours)

**Figure 6.** The effects of shot numbers on the test performances of different methods are reported with 95% confidence intervals.

Although it can solve supervision collapse in small sample scene classification, our proposed method has no advantage in terms of time complexity. The time complexity of SSMR is dominated by the self-supervised learning module, meta-learning module, and self-distillation module. The self-supervised learning module needs to extract two sets of local features and global features for cross comparison. The self-supervised learning module has a time complexity of $M^2$; the time complexity of the meta-learning module for metric classification is $N$; and the time complexity of the self-distillation model is determined by the number of iterations $Z$. The total time complexity of our proposed method is $O(M^2 + N + Z)$. In the experiment, we trained 100 epochs and it took 19.6 h to train using a single NVIDIA GeForce RTX 3090 GPU. Among the methods with similar classification performance to SSMR under the same experimental conditions, it takes 15.5 h to train RENET, 11.3 h to train ProtoNet, and 12.2 h to train MAML. As can be seen, our method achieves better performance but requires more computational resources. This is one of the shortcomings of our proposed method.

*4.5. Feature Maps*

We loaded the trained model, fed a randomly selected remote sensing image into the model, and extracted two sets of feature maps from the embedding network, as shown in Figure 7. The first set of feature maps is from the intermediate layer of the embedding

network, and the second set of feature maps is from the end of the embedding network. It can be seen that the composition of the features extracted from the embedding network is very complex. The embedding network extracts not only the features of the key targets in the input image, but also other features. This is due to the fact that the features extracted by the trained embedding network contain both local features and global features in the input image. Moreover, the embedding network wants to set this remote sensing image as far as possible from other categories (including natural categories) of data in the feature metric space, so it also extracts additional features that are helpful to achieve this goal and finally extracts feature maps containing complex feature information.

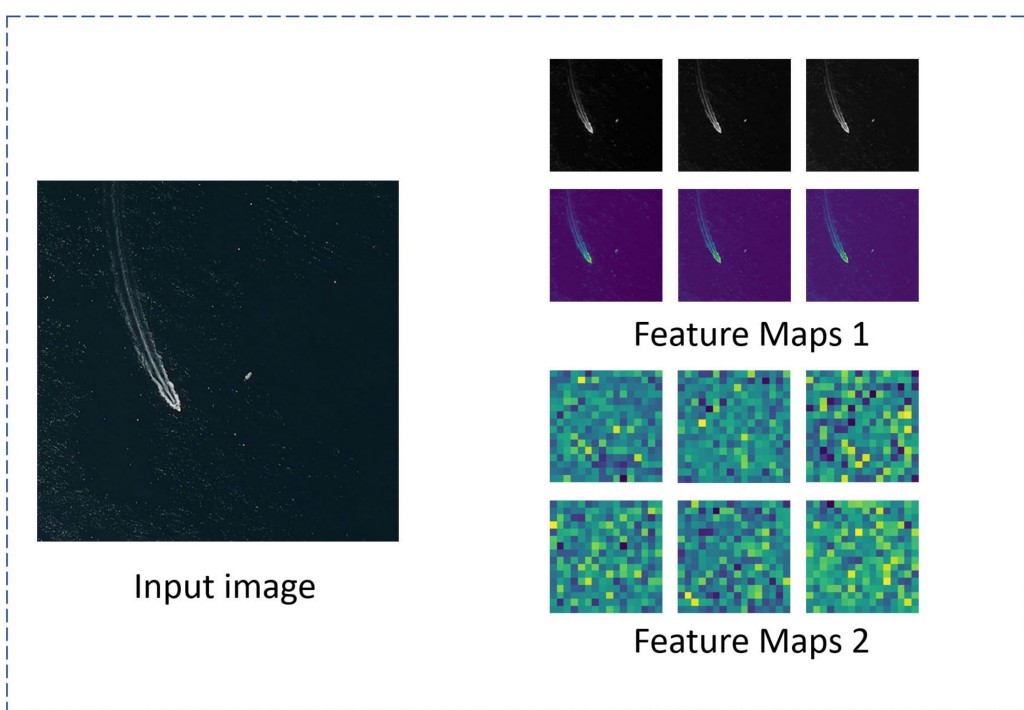

**Figure 7.** A randomly selected remote sensing image and two sets of feature maps extracted from the embedding network.

### 4.6. Ablation Experiment

With ablation experiments, we examined the impact of different choices made in each structure of the proposed method on the performance of the method. We also verified the effectiveness of the self-supervised learning part and the self-distillation part through ablation experiments.

#### 4.6.1. Selection of Contrastive Learning Features

When selecting the object of contrastive learning in self-supervised learning in Section 3.1, the local features of different sources will influence the effect of self-supervised learning and further affect the performance of the scene classification model. We chose $1 \times 1$ local features, $5 \times 5$ local features, and $7 \times 7$ local features to compare the local features generated by three different positions and global features. During model training, other parameter settings are the same in the three groups of experiments, and then the performance of the model is tested. Table 5 illustrates the results. Selecting $7 \times 7$ local features, the performance of the model is the best, followed by $5 \times 5$ local features, and finally $1 \times 1$ local features. We think this is because $7 \times 7$ local features contain more feature information for contrastive learning, while $1 \times 1$ local features are too abstract for global features such that the difficulty of contrastive learning increases and the mutual information that can be learned decreases.

**Table 5.** Analysis of different features chosen for contrastive learning effects on one-shot five-way vs. five-shot five-way task accuracy (%). There is a 95% confidence interval associated with all results.

| Local Feature | One-Shot Five-Way Accuracy | Five-Shot Five-Way Accuracy |
|:---:|:---:|:---:|
| $1 \times 1$ | $64.33 \pm 0.20$ | $81.67 \pm 0.14$ |
| $5 \times 5$ | $65.17 \pm 0.15$ | $82.14 \pm 0.16$ |
| $7 \times 7$ | $66.52 \pm 0.20$ | $83.26 \pm 0.49$ |

### 4.6.2. Data Augmentation

A comparison was made between different methods of data augmentation and how they affected the performance of the method. A total of three groups of experiments were set up, using resize and CenterCrop; ColorJitter to adjust brightness, contrast, saturation and hue; and transforming remote sensing images into gray images randomly. We also used two new data augmentation methods proposed in recent years. RandAugment [70] is an automated data augmentation strategy that does not require a separate search. It can employ a simple grid search to generate a rich number of image transformations. It has achieved very powerful results in many datasets. The data augmentation strategy used in SimCLR [45] is widely used because of its excellent effect. It includes: resize and random horizontal flip; uses ColorJitter to adjust brightness, contrast, saturation and hue; and optional grayscale conversion. Finally, it uses Gaussian blur and solarization. Using these two powerful data augmentation methods instead of the simple methods usually leads to significant performance improvements. In Table 6, you can see the results of the experiment.

**Table 6.** Comparison of the accuracy (%) of a five-way task when performed with five shots and one shot with different data augmentation. We report all results with a 95% confidence interval.

| Data Augmentation | One-Shot Five-Way Accuracy | Five-Shot Five-Way Accuracy |
|:---:|:---:|:---:|
| ColorJitter | $64.76 \pm 0.15$ | $81.83 \pm 0.30$ |
| RandomGrayscale | $64.60 \pm 0.23$ | $82.38 \pm 0.45$ |
| Resize + CenterCrop | $65.33 \pm 0.20$ | $82.67 \pm 0.14$ |
| Strategy in SimCLR [45] | $63.81 \pm 0.43$ | $80.19 \pm 0.67$ |
| RandAugment [70] | $62.59 \pm 0.68$ | $79.83 \pm 0.11$ |

Comparing the results of four groups of experiments, the best effect of the three data augmentation methods is achieved with resize and CenterCrop. We believe that this is related to the image obtained with remote sensing having characteristics of high spatial resolution and spectral resolution, such that the convolution network may rely more on texture and shape in the classification of remote sensing image scenes. Therefore, the data augmentation transforming the shape of remote sensing images is better. Surprisingly, data augmentation using RandAugment and strategy in SimCLR did not lead to more performance gains, but rather to a lesser performance boost. With regard to this counter-intuitive phenomenon, we believe that the overly aggressive data augmentation strategy damages the original data distribution pattern in the few-shot dataset and reduces the signal-to-noise ratio in the few-shot dataset. The number of natural category samples in our dataset is larger than the number of remote sensing image samples. According to [71], the powerful data augmentation strategy is biased towards the natural category samples, which in turn deepens the imbalance between categories and ultimately leads to poorer performance than expected.

### 4.6.3. Optimizer

The performance of different optimizers is different in different scenarios. During the experiment, we explored the influence of optimizer type on the performance of the model. We exercised the model with the same structure with SGD and Adam optimizers, respectively, and tested it in five-shot five-way and one-shot five-way tasks

(Table 7). According to the comparison, SGD is more suitable for the method proposed in this paper. We believe that adaptive learning rate algorithms such as Adam have advantages in fast convergence. However, few-shot data belong to the sparse data. The statistical characteristics of the sparse data are so poor that the error surface in the training process is complex. SGD (+momentum) with fine-tuning parameters can finally achieve better results.

**Table 7.** Comparison of the accuracy (%) of the five-way task when performed with five shots and one shot (%) between SGD and Adam optimizer. Ninety-five percent confidence intervals are provided for all results.

| Optimizer | One-Shot Five-Way Accuracy | Five-Shot Five-Way Accuracy |
|---|---|---|
| Adam [67] | $59.67 \pm 0.19$ | $78.04 \pm 0.35$ |
| SGD [25] | $66.52 \pm 0.20$ | $83.26 \pm 0.49$ |

4.6.4. Self-Supervised Learning Ablation Experiment

A number of experiments have been conducted to confirm the effectiveness of self-supervised learning. We substituted the embedding network trained by self-supervised learning in the model with ResNet18 and ConvNet, and other structures and parameters remained unchanged. After replacing the embedding network, the network was trained by supervised learning rather than self-supervised learning. The comparison results for one-shot and five-shot five-way tasks are shown in Table 8. After using supervised learning instead of self-supervised learning, the performance of the model decreased significantly. This was because supervised learning only uses label information to train the embedding network and overfitting occurs when too few samples are collected, showing that using self-supervised learning can extract supervision other than label information and improve the performance of embedding networks.

**Table 8.** Analysis of several embedding networks' effects on one-shot five-way vs. five-shot five-way task accuracy (%). Ninety-five percent confidence intervals are given for all results.

| Embedding Network | One-Shot Five-Way Accuracy | Five-Shot Five-Way Accuracy |
|---|---|---|
| ConvNet | $34.50 \pm 0.12$ | $40.59 \pm 0.35$ |
| ResNet18 | $49.33 \pm 0.26$ | $62.77 \pm 0.51$ |
| Self-supervised network | $66.52 \pm 0.20$ | $83.26 \pm 0.49$ |

4.6.5. Self-Distillation Ablation Experiment

We also performed ablation experiments on the self-distillation part. In the same dataset, the classification model is only trained in self-supervised learning and meta-learning stages without self-distillation training. We studied the performance of the obtained model with the SSMR method in the one-shot five-way scenario and the five-shot five-way scenario, as shown in Table 9. The method with additional self-distillation training performed better in few-shot classification tasks. The self-distillation module can provide 0.9–1.3% accuracy improvement.

**Table 9.** A study of the accuracy (%) of the five-way task when performed with five shots and one shot between training with self-distillation or withouut. All average results are shown with 95% confidence intervals.

| Method | One-Shot Five-Way Accuracy | Five-Shot Five-Way Accuracy |
|---|---|---|
| SMR (no self-distillation) | $65.61 \pm 0.56$ | $81.91 \pm 0.27$ |
| SSMR | $66.52 \pm 0.20$ | $83.26 \pm 0.49$ |

## 5. Conclusions

Meta-learning has a crucial role to play in few-shot remote sensing image scene classification. However, this work shows that there is a hidden problem named supervision collapse: the ways of embedding networks composing features are often so eager to learn invariances that information about intra-class variation that may be necessary to understand novel classes is lost. The supervision collapse in remote sensing image studies has not received much attention, and we have tried to call attention to the existence of this problem. We have proposed a method named SSMR which can overcome supervision collapse and complete scene classification in few-shot remote sensing image scenarios well without additional data annotation. Firstly, self-supervised learning with multi-layer feature contrast was constructed to train an embedding network with excellent performance for meta-learning. This embedding network can make use of the general and effective expression extracted from the training task to reduce the impact of supervision collapse on the effect of meta-learning for better completion of cross-domain tasks. Secondly, multi-layer feature information and category label information were combined to guide self-distillation for more in-depth model optimization. Experiments have shown that our method overcomes supervision collapse and performs better than previous methods in challenging tasks. Every coin has two sides, and our proposed method also has some weaknesses. Firstly, the proposed method has high time complexity; the training process needs a lot of time and computing resources. Secondly, the structure of our proposed method is not a standard end-to-end structure, and it is difficult to transplant to resource-constrained environments (such as embedded devices). Combined with the first point, there are difficulties associated with the application of the algorithm in mobile edge computing devices. Thirdly, unlike the scene classification models built using the ConvNet and ResNet structures, which can be easily migrated to remote sensing interpretation tasks as classifiers, the proposed method is difficult to directly migrate to remote sensing interpretation tasks because it contains special designs for supervision collapse. In future work, we will look to improve our method and address these weaknesses. We can choose a more simple and efficient self-supervised learning paradigm to train the embedding network to reduce the time complexity of the method, integrate the different modules of the method into a standard end-to-end structure and use easily quantifiable modules to realize the embedded transplantation of the method, and explore ways to apply the model in other remote sensing image interpretation tasks. For example, we could use the features extracted by the embedding network for remote sensing image captioning and remote sensing image segmentation.

**Author Contributions:** Conceptualization, H.Z.; methodology, H.Z.; software, H.Z.; validation, H.Z.; formal analysis, H.Z.; investigation, S.L.; resources, X.D.; data curation, H.Z.; writing—original draft preparation, H.Z.; writing—review and editing, H.Z.; visualization, H.Z.; supervision, X.D.; project administration, H.Z. All authors have read and agreed to the published version of the manuscript.

**Funding:** This research received no external funding.

**Data Availability Statement:** The MASATI dataset can be obtained at https://www.iuii.ua.es/datasets/masati/index.html (accessed on 15 April 2022). The Miniimagenet dataset can be obtained at https://lyy.mpi-inf.mpg.de/mtl/download/Lmzjm9tX.html (accessed on 15 April 2022).

**Conflicts of Interest:** The authors declare no conflict of interest.

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
