# Peer review of "Self-Supervision and Self-Distillation with Multilayer Feature Contrast for Supervision Collapse in Few-Shot Remote Sensing Scene Classification"

_remotesensing, doi:10.3390/rs14133111_

Round 1
Reviewer 1 Report
The paper proposes a solution for the scene classification task. The idea is based on conv nets and augmentation methods. The model is described with all necessary re-implementing details, but some issues are missing. Therefore, i must ask the authors for adding this information and making some discussion on some parts of their proposal like:
1)Underline the novelty in the abstract
2)Discuss the use of other machine learning models in the ship classification, the monitoring of coastal, etc.
3)Discuss also other, hybrid models based on machine learning like neural bag-of-words.
4)Explain why the new architecture is needed when we have so many pre-trained models. I understand, that your solution is based on such a model (embedding net), but why such a network is not enough.
5)Did you test some other augmentation tools in your proposal? The last few years brought some interpolation merges, etc. Some discussions also could be worth making.
6)Some information about the kernel sizes and their selection methods should be added. Can you show the feature maps of these kernels?
7) The loss function is based on sum of other ones (see Eq. (2)) - why are all of them equal?
8) Experimental section is well-designed.
Reviewer 2 Report
1. Figure 1 that representing the overall structure is unclear, especially for the multi-layer feature contrast process.
2. In part 4.1, number and categories of the mixed dataset are not clear enough, a table is better provided.
3. In Table 1, structure of the network is not supposed to be right, where the FC layer is absent. Also, the last output should not be 1x1 vector for your classification model.
4. In Line 553-555, there is no evidence to support the statement that low performance is due to the self-attention mechanism (Transformer). You may need to change the structure of Embedding-Net (ConvNet) of your method with Transformer to support this idea.
5. The novelty is not clear and most of the techniques uses are well known approaches.
6. The listed algorithm/method should be formalized. The corresponding time complexity should be analyzed.
7. Discuss the limitations of the proposed method. Every method has a weakness and should be discussed clearly and give solutions in the future work section.
8. The whole text should be carefully examined and corrected. Lots of language error. Looks like they have not edited the paper
Typo and Language error
1. Lines 111 “reduce the compactness …”
2. Lines 239, have no idea of “niece NN”
3. In Figure 381, “will over cluster learning”
Round 2
Reviewer 1 Report
The paper was improved to all my comments.